# MEMORY-AUGMENTED DESIGN OF GRAPH NEURAL NETWORKS

## ABSTRACT

The expressive power of graph neural networks (GNN) has drawn much interest recently. Most existent work focused on measuring the expressiveness of GNN through the task of distinguishing between graphs. In this paper, we inspect the representation limits of locally unordered messaging passing (LUMP) GNN architecture through the lens of *node classification*. For GNNs based on permutation invariant local aggregators, we characterize graph-theoretic conditions under which such GNNs fail to discriminate simple instances, regardless of underlying architecture or network depth. To overcome this limitation, we propose a novel framework to augment GNNs with global graph information called *memory augmentation*. Specifically, we allow every node in the original graph to interact with a group of memory nodes. For each node, information from all the other nodes in the graph can be gleaned through the relay of the memory nodes. For proper backbone architectures like GAT and GCN, memory augmented GNNs are theoretically shown to be more expressive than LUMP GNNs. Empirical evaluations demonstrate the significant improvement of memory augmentation. In particular, memory augmented GAT and GCN are shown to either outperform or closely match state-of-the-art performance across various benchmark datasets.

## 1 INTRODUCTION

Graph neural networks (GNN) are a powerful tool for learning with graph-structured data, and has achieved great success on problems like node classification (Kipf & Welling, 2016), graph classification (Duvenaud et al., 2015) and link prediction (Grover & Leskovec, 2016). GNNs typically follow a recursive neighborhood aggregation (or message passing) scheme (Xu et al., 2019) such that within each aggregation step, each node collects information from its neighborhood (usually feature vectors), then apply aggregation and combination mechanism to compute its new feature vector. Typically, GNN architectures differ in their design of aggregation and combination mechanisms. Popular architectures like GCN (Kipf & Welling, 2016), GraphSAGE (Hamilton et al., 2017), and GAT (Veličković et al., 2018) fall into this paradigm.

Despite their empirical success, there are a couple of limitations of GNNs that update node features only based on local information. One important issue is their limited expressive power. In the graph classification setting (Xu et al., 2019), it was shown that message passing neural networks are at most as powerful as Weisfeiler Lehman graph isomorphism tests. A more recent line of work has suggested using variants of message passing scheme that incorporates the layout of local neighborhoods (Sato et al., 2019; Klicpera et al., 2020) or spatial information of the graph (You et al., 2019).

Another problem is due to the phenomenon that the performance of GNN does not improve, or even degrades when layer size increases (Kipf & Welling, 2016; Xu et al., 2018; Li et al., 2018; Oono & Suzuki, 2020), known as the problem of *over-smoothing* that makes extending the receptive path of message passing GNNs a difficult task. Many successful GNN architectures are based on stacking a few number of layers like 2 or 3 (Kipf & Welling, 2016), which could be viewed as an implicit inductive bias that node labels are determined up to neighborhoods that are a few hops away. However this assumption may not hold for many real-world data—for example, structurally similar nodes may offer strong predictive power for very distant node pairs (Donnat et al., 2018). Several techniques are proposed for aggregating node information from a wider range (Xu et al., 2018; Klicpera et al., 2019a;b).

In this paper, we investigate the expressive power of GNNs through the task of node classification. We characterize cases where GNNs that builds on LUMP protocol fail, regardless of underlying implementation or aggregation range. We then propose a novel architecture that aggregates information beyond the local neighborhood. Through making use of global feature information, we can distinguish a wide range of cases that LUMP type GNNs inherently fail. Our main contributions are summarized as follows:

- We discuss the expressive power of GNNs for node classification tasks under the premise that node labels are not solely determined by first-order neighborhood information, and show an indistinguishable scenario where LUMP algorithms fail to discriminate nodes in structurally different graphs even if infinite rounds of message passing is performed.
- We develop a novel framework that extends GNN with *global graph information* called *memory augmentation*, motivated by memory networks (Graves et al., 2014; Weston et al., 2014). With proper choice of backbone architectures like GAT or GCN, the augmented architectures are provably more expressive than LUMP type GNNs in that they discriminate a wide range of cases that LUMP type GNNs fail with a compact architecture of two layers.
- We derive two representative memory augmented architectures, MemGAT and MemGCN, and evaluate their performance on standard datasets. Empirical results show that the memory augmented architectures significantly improves their corresponding backbone architectures across all tasks, either outperforming or closely matching state-of-the-art performance.

## 2 REPRESENTATION LIMITS OF LOCALLY UNORDERED MESSAGE PASSING

In this paper we consider the task of *node classification* over an undirected graph $G = (V, E)$ with node set $V$ and edge set $E$. Let $N = |V|$ be the number of nodes and $A, D$ be its associated adjacency matrix and degree matrix. For each node $v \in V$, let $\mathcal{N}_v = \{u \mid (u, v) \in E\}$ be its neighborhood set and $X_v \in \mathcal{X} \subset \mathbb{R}^d$ be its associating feature. Each node $v \in V$ is associated with a label $Y_v$. Node classification algorithms make predictions of $Y_v$ based on the information given by $G$ and the node feature matrix $X$. In this paper we will be interested in situations where node labels are not determined solely by their first order neighborhood information. i.e., $\mathbb{P}(Y_v | G, X) \neq \mathbb{P}(Y_v | X_v, X_u, u \in \mathcal{N}_v), \forall v \in V$. For a collection of elements $\mathcal{C}$ that are not necessarily distinct, we use $\{\mathcal{C}\}$ to denote its set representation and $\{\!\{\mathcal{C}\}\!\}$ to denote its multiset representation. For each $c \in \{\mathcal{C}\}$, let $r_\mathcal{C}(c)$ be the multiplicity of $c$ in $\{\!\{\mathcal{C}\}\!\}$. A popular tool for encoding higher order graph information is to utilize the *locally unordered message passing (LUMP)* protocol (Garg et al., 2020) to build GNNs. For node $v$, its (hidden) representation $h_v^{(l)}$ is updated using an *aggregation and combine* strategy:

$$h_v^{(l)} = \mathsf{COMBINE}\left(h_v^{(l-1)}, \mathsf{AGG}\left(\left\{\!\!\left\{h_u^{(l-1)}, u \in \mathcal{N}_v\right\}\!\!\right\}\right)\right) \tag{1}$$

The protocol is unordered in the sense that no spatial information (like the relative orientation of neighbors) is used throughout the message passing procedure, and the aggregator $\mathsf{AGG}$ is often chosen as a permutation invariant function. After $k$ rounds of message passing, each node will have a feature vector that encodes the information of its *height $k$ rooted subtree*. Aggregation strategies that extend to arbitrary node were suggested in pioneering works of GNNs (Scarselli et al., 2009) that use a learnable, *contractive* aggregator, and perform infinite rounds of message passing till convergence.

Next we discuss the expressive power of the above mentioned mechanisms. Let $\mathsf{G}(v)$ be the subgraph of $G$ that contains $v, \mathcal{N}_v$ and their associated edges. We consider two graphs, $G = (V, E)$ and $G' = (V', E')$, with corresponding feature matrices $X$ and $X'$.

**Definition 1.** (Gross & Tucker, 2001) A graph map $f : G \mapsto G'$ is called a *local isomorphism* if for every node $v \in V$, the restriction of $f$ to $\mathsf{G}(v)$ is an isomorphism onto $\mathsf{G}(f(v))$.

Local isomorphism could be understood as a relaxed version of graph isomorphism. In particular, for two isomorphic graphs, the isomorphism map is also a local isomorphism but the converse is not true (see figure 1). Next we use the notion of local isomorphism to help characterize the expressive power of GNNs in node classification context. We say a graph $G$ is *locally indistinguishable* to graph $G'$, if there exists a surjective local isomorphism $f$ from $G$ to $G'$, and if in addition the feature matrices are related as $X_v = X'_{f(v)}$. The following theorem states a specific situation where LUMP type GNNs fail to distinguish between nodes in different graph contexts.

**Theorem 1.** *If $G$ is locally indistinguishable to graph $G'$ under map $f$, then under the LUMP protocol equation 1, we have $h_v^{(l)} = h_{f(v)}^{(l)'}$ for any $v \in V$ and $l \in \mathbb{Z}_+$.*

The proof will be given in appendix A.2. We give a pictorial illustration in figure 1, with node color indicating feature value. The local isomorphism is constructed as $f(a_1) = f(a_4) = b_1, f(a_2) = f(a_5) = b_2, f(a_3) = f(a_6) = b_3$. Theorem 1 implies that representations of nodes with same color remain identical for arbitrary rounds of LUMP. Hence for the node label assignment $l_{a_1} = 1, l_{b_1} = 0$, GNNs based on LUMP fail to express this difference.

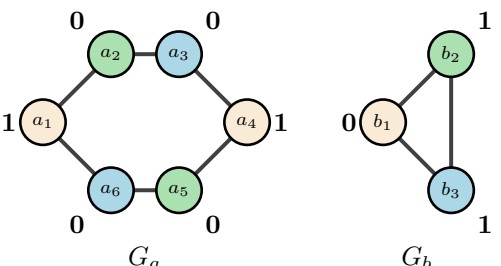

Figure 1: Graph structures that LUMP GNNs fail to discriminate in terms of node classification tasks. (with respect to the assignment annotated near the corresponding node)

**Remark 1.** *Figure 1 is a contrived case that seldom happens in practice. Note that such cases correspond to the worst case under which* infinite *depth LUMP architecture fails. For a more realistic setup, we may consider relaxing the surjectivity requirement in the definition of local indistinguishable graphs: if two nodes from different graphs have their induced $k$-hop neighborhood being isomorphic, it follows that any depth-$k$ LUMP architectures fail to distinguish these two nodes. Locally isomorphic subgraphs are frequently observed in chemistry and bioinformatics (Raymond & Willett, 2002; Cao et al., 2008). Although locally isomorphic subgraphs from globally distinct graphs may get discriminated via using deep LUMP architectures, it is non-trivial to scale GNNs to deep architectures without losing expressivity due to the phenomenon of over-smoothing Xu et al. (2018).*

## 3 OUR MODEL

### 3.1 MEMORY AUGMENTATION

Our model is constructed via augmenting a *backbone GNN* whose updating rule could be written as a simplified version of equation 1: the hidden representation in the $l$th layer satisfies $h_v^{(l+1)} = \mathsf{AGG}_{\vartheta^{(l)}} \left( \left\{ h_v^{(l)}, h_{\mathcal{N}_v}^{(l)} \right\} \right)$. The aggregation function $\mathsf{AGG}$ is parameterized by $\vartheta^{(l)}$. We will assume the set $\mathcal{X}$ to be countable, then by (Xu et al., 2019, Lemma 4), the range of $h_v^{(l)}$, denoted $\mathcal{H}^{(l)}$, is also countable for any given parameterization and any layer $l$.

**Augmentation** Inspired by memory networks (Graves et al., 2014; Weston et al., 2014), the memory component is formulated as a collection of learnable objects corresponding to $M \ll N$ auxiliary nodes, called *memory nodes*. Let $V_{\mathcal{M}}$ be the set of memory nodes. We define memory as an $M \times d$ matrix $\mathbf{m}$ with row vector $\mathbf{m}_v$ serving as the *memory embedding* of memory node $v$. We allow every node in the original graph to interact with the memory component. In particular, we augment the original graph $G$ into a slightly larger graph $G^* = (V^*, E^*)$, where $V^* = V \bigcup V_{\mathcal{M}}$ and

$$E_{ij}^* = \begin{cases} E_{ij}, & \text{if } i, j \in V \\ 1, & \text{if } i \in V, j \in V_{\mathcal{M}} \text{ or } i \in V_{\mathcal{M}}, j \in V \\ 0, & \text{if } i, j \in V_{\mathcal{M}} \end{cases} \tag{2}$$

so that each memory node is connected to every node in the original graph $G$. Performing message passing using the backbone architecture would thus allow every node in the original graph to aggregate information from both its neighbors and the memory component, while the memory component aggregates information from the whole graph:

$$\forall v \in V, h_v^{(l+1)} = \mathsf{AGG}_{\vartheta^{(l)}} \left( \left\{ h_v^{(l)}, h_{\mathcal{N}_v}^{(l)}, \mathbf{m}_{V_{\mathcal{M}}} \right\} \right)$$

$$\forall v \in V_{\mathcal{M}}, \mathbf{m}_v^{(l+1)} = \mathsf{AGG}_{\vartheta^{(l)}} \left( \left\{ \mathbf{m}_v^l, h_V^{(l)} \right\} \right)$$

Figure 2 illustrates how memory augmentation does "symmetry breaking" in locally indistinguishable graphs: although the height $k$ rooted subtree remains identical for $G_a$ and $G_b$ for any $k$, starting

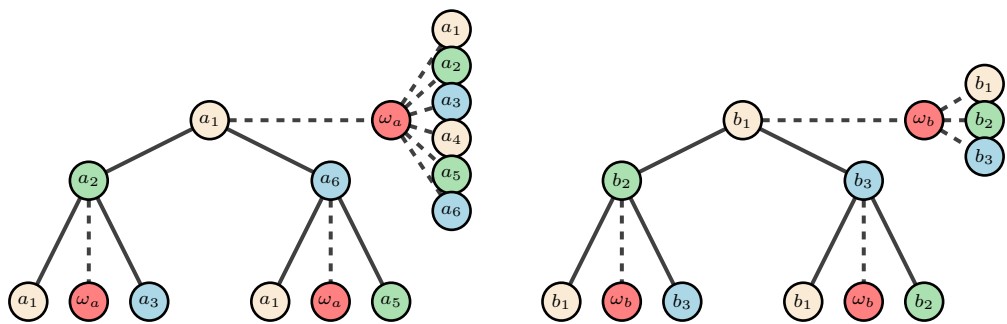

Figure 2: Illustration of memory augmentation applied to the two graphs in figure 1: $G_a$ (left) and $G_b$ (right). Both graphs are augmented with one memory node. Aggregation paths of $a_1$ and $b_1$ are presented as two hop subtree structures, corresponding to a two round message passing using the memory augmented attention mechanism. Memory connections are drawn as dashed lines and original connections as solid lines. The figure implies that, although the local subtree structures rooted at $a_1$ and $b_1$ are identical, the message carried by the memory node distinguishes $a_1$ from $b_1$, provided that the message passing is performed for more than one round.

from the second layer, messages from the memory component would become distinct thus makes discrimination possible. More rigorously, we characterize conditions for the backbone architecture under which the memory augmented architecture achieves stronger expressive power:

**Definition 2.** We say two multisets $\{X\}$ and $\{X'\}$ are distributionally equivalent if they share the same underlying set, i.e. $\{X\} = \{X'\}$, and the corresponding multiplicities are proportional to each other: for $\forall x \in \{X\}, r_X(x)/r_{X'}(x) \equiv m, m \in \mathbb{Z}_+$.

Note that distributional equivalence is a well defined equivalence relation. We require the following condition on the backbone GNN:

[$\mathbf{C}_1$] The backbone GNN identifies bounded size multisets in $\mathcal{X}$ up to distributional equivalence.

Using a single memory node $\omega$, we have the following result:

**Lemma 1.** *Under condition $\mathbf{C}_1$, there exists a vector $\tilde{\mathbf{m}}_\omega$ such that if the memory embedding is initialized with $\mathbf{m}_{\omega,G}^{(0)} = \tilde{\mathbf{m}}_\omega$ for any graph $G$, then the following holds:*

*(i) For any $l \in \mathbb{Z}_+$, there exists a parameter configuration $\vartheta_*^{(l)}$ such that the map $\mathsf{AGG}_{\vartheta_*^{(l)}}$ is injective over all multisets that takes the form: $\left\{C, \mathbf{m}_{\omega,G}^{(l)}\right\}$, where $\{C\}$ is a bounded subset of $\mathcal{X}$*

*(ii) For graph pairs $(G, G')$ with $\{X\} \neq \{X'\}$, the updated memory representations after 1 round of message passing are different, i.e. $\mathbf{m}_{\omega,G}^{(1)} \neq \mathbf{m}_{\omega,G'}^{(1)}$.*

It then follows from (Xu et al., 2019, Theorem 3) that memory augmentation enhances proper backbone architectures to achieve the strongest expressive power among LUMP GNNs. Lemma 1 suggests that as long as the overall network architecture contains more than one layer, starting from the second layer, nodes in the original graph would receive messages that contain some kind of *aggregated global information* that *identifies* the multiset feature representation of the original graph. This could be also viewed as a *graph specific bias* term that discriminates that message passing protocol from LUMP. As a consequence, we have the following theorem for binary node labels:

**Theorem 2.** *For locally indistinguishable graphs $(G, G')$ under map $f$, there exists a two layer memory augmented GNN with its backbone architecture satisfying condition $\mathbf{C}_1$, such that with a proper memory initialization (i.e. the one in lemma 1), the final output $h$ satisfies: if $\{X\} \neq \{X'\}$, then $h_v > 0.5$ and $h_{f(v)} \leq 0.5$ for any $v \in V$.*

Theorem 2 suggests that memory augmentation helps to identify locally indistinguishable graphs with different multiset feature representations. The following corollary justifies using two most popular

GNN architecture as the backbone architecture for memory augmentations. We adopt the formulation in Xu et al. (2019):

**Corollary 1.** Both GAT and GCN satisfies condition $\mathbf{C}_1$.

We defer all proofs to appendix A.2. For an empirical verification, we conducted a simple experiment corresponding to the setup in figure 1. We use categorical features (with a cardinality of three) and compared memory augmented GAT with GAT. We use the reference GAT architecture in (Veličković et al., 2018) and augment it with one memory node. We use a learnable multi-layer perceptron as the nonlinear readout function for both models. Figure 3 shows the result. It could be seen from the figure that the training accuracy of GAT never exceeds 2/3, which is its theoretical limit in this example. While memory augmented GAT fits the data perfectly after sufficient rounds of gradient updates, thereby verifies our theoretical findings.

## 3.2 TWO CANONICAL DESIGNS: MEMGAT AND MEMGCN

Now we derive two canonical architectures that enhances two popular GNNs GAT (Veličković et al., 2018) and GCN (Kipf & Welling, 2016). The resulting architectures are termed MemGAT and MemGCN. For both designs, we introduce several improvements to the vanilla memory augmentation described in section 3.1 to balance the contribution from the original graph and the memory component. We also make a discussion on architectures that may not benefits from memory augmentation, see appendix A.4 for details.

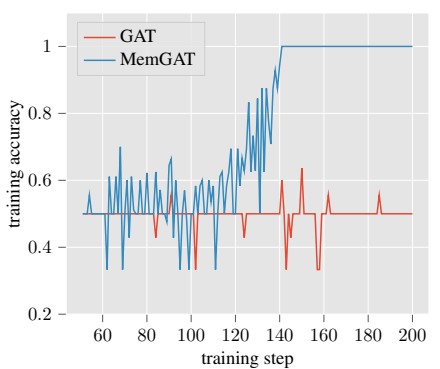

**MemGAT** We enhance attention-based GNNs (Veličković et al., 2018) via using a modified attention mechanism. The attention weights are calculated as:

$$
\alpha_{ij}^{(l)} = \begin{cases}
\lambda \psi_{ij} \exp\left(\beta_{ij}^{(l)}\right)/Z_i^{(l)}, & \text{if } i,j \in V \\
\frac{1-\lambda}{M} \exp\left(\beta_{ij}^{(l)}\right)/Z_i^{(l)}, & \text{if } i \in V, j \in V_{\mathcal{M}} \\
\exp\left(\beta_{ij}^{(l)}\right)/Y_i^{(l)}, & \text{if } i \in V_{\mathcal{M}}, j \in V \\
\mathbf{1}_{(i=j)}\exp\left(\beta_{ij}^{(l)}\right)/Y_i^{(l)} & \text{if } i,j \in V_{\mathcal{M}}
\end{cases}
\tag{3}
$$

Figure 3: Comparison of memory augmented GAT with GAT on synthesized data, performance measured in training accuracy, results for initial 50 gradient iterations are discarded

where $h_i^{(l)} \in \mathbb{R}^{d_l}$ denotes the hidden feature of node $i \in V$ in the $l$ th layer with $h_i^{(0)} = X_i$, and $\beta_{ij}^{(l)} = \phi^{(l)}(\sigma^{(l)}(h_i^{(l)}), \sigma^{(l)}(h_j^{(l)}))$ with $\sigma^{(l)}$ a (possibly) nonlinear function for the $l$ th layer. $\{\psi_{ij}\}_{i,j,\in V}$ are edge weights over the original graph, and $\lambda \in (0,1)$ is a parameter that balances the contribution between information provided by the original neighbor and the memory component. $\{Z_i^{(l)}\}_{i\in V}, \{Y_j^{(l)}\}_{j\in V_{\mathcal{M}}}$ are normalizing factors such that $\sum_j \alpha_{ij}^{(l)} = 1, \forall l$. We write separately the updating equation for hidden representations corresponding to nodes in the original graph and memory embeddings:

$$
\forall v \in V, \quad h_i^{(l+1)} = \sum_{k \in \{i\}\bigcup \mathcal{N}_i} \alpha_{ik}^{(l)}\sigma^{(l)}(h_k^{(l)}) + \sum_{v \in V_{\mathcal{M}}} \alpha_{iv}^{(l)}\sigma^{(l)}(\mathbf{m_v}^{(l)}) \tag{READ}
$$

$$
\forall v \in V_{\mathcal{M}}, \quad \mathbf{m_v}^{(l+1)} = \alpha_{vv}^{(l)}\sigma^{(l)}(\mathbf{m_v}^{(l)}) + \sum_{j\in V} \alpha_{vj}^{(l)}\sigma^{(l)}(h_j^{(l)}) \tag{WRITE}
$$

**Connections to memory based network design** The above equations interprets the memory component as content addressing memory design in memory-based neural networks (Graves et al., 2014; Weston et al., 2014): For nodes in the original graph, the aggregation step is interpreted as a *reading* step with a selective focus over both its immediate neighbors and the memory obtained from the previous layer. Alternatively, for the memory nodes, aggregation operations are hence interpreted as a *writing* step.

**MemGCN**  The augmented GCN adopts similar design aspects as MemGAT with balancing parameter $\lambda$. We adopt the GCN formulation in Dehmamy et al. (2019), by using edge weights $\{\psi_{ij}\}_{i,j,\in V}$ to characterize the aggregation scheme:

$$\forall v \in V, \quad h_i^{(l+1)} = \lambda \left( \psi_{ii} W^{(l)} \sigma^{(l)}(h_i^{(l)}) + \sum_{j \in \mathcal{N}_i} \psi_{ij} W^{(l)} \sigma^{(l)}(h_i^{(l)}) \right) + \frac{1-\lambda}{M} \sum_{v \in V_{\mathcal{M}}} W \mathbf{m}_v^{(l)}$$

$$\forall v \in V_{\mathcal{M}}, \quad \mathbf{m_v}^{(l+1)} = \lambda W^{(l)} \sigma^{(l)}(\mathbf{m_v}^{(l)}) + \frac{1-\lambda}{M} \sum_{j \in V} W^{(l)} \sigma^{(l)}(h_i^{(l)})$$

where $W^{(l)}$s are learnable weight matrices. We identify two canonical types of GCN: Kipf & Welling (2016) uses $\psi_{ij} = 1/\sqrt{(d_i+1)(d_j+1)}$, and a slightly modified method of Hamilton et al. (2017) uses $\psi_{ij} = 1/(d_i+1)$.

**Remark 2.** *Recently several variants of LUMP that take ordered information into account are proposed, among which CPCGNN (Sato et al., 2019) and DimeNet (Klicpera et al., 2020) are based on modifications to the first-order message passing scheme, and are provably more expressive than LUMP type GNNs (Garg et al., 2020). We give a detailed comparison of memory augmented GNN architectures with CPCGNN and DimeNet in appendix A.5. We found that regarding node classification, there's no dominating choice between these methods. And it would be of interest to extend the idea of memory augmentation to other kinds of message passing schemes. We leave this direction to future research.*

## 4    OTHER RELATED WORK

The expressive power of GNNs has been extensively studied recently (Xu et al., 2019; Maron et al., 2019; Garg et al., 2020; Sato et al., 2020; Zhang & Xie, 2020), with the majority of these works defining the expressive power of GNN with its ability to distinguish different graphs. (Xu et al., 2019) proposed the GIN model that is theoretically as powerful as the first order Weisfeiler Lehman (1-WL) test and is thus the most powerful among LUMP GNNs. (Zhang & Xie, 2020) suggests accounting for cardinality information for attention based GNNs. More powerful GNN architectures are then proposed via extending the message passing formulation: higher order GNNs or $k$-GNNs (Morris et al., 2019; Maron et al., 2019) operate on $k$- tuple of nodes and aggregate information from neighborhoods defined on $k$- tuples of nodes. While such kind of GNN variants could be shown to be as powerful as $k$-WL test, they are not directly applicable to node level tasks. Another line of work (Sato et al., 2019; Klicpera et al., 2020) generalizes the LUMP protocol to allow nodes passing messages depending on their relative position in their belonging neighborhoods. Such kind of variants are also provably more powerful than LUMP GNNs and are applicable to node level tasks. See section 6 for a more detailed comparison between our approach and these variants. Position aware GNN (PGNN) (You et al., 2019) is another class of powerful GNN variants that uses random anchor sets to break local structural symmetry, and integrates positional information into a modified message passing architecture, with an inductive bias suitable for relational classification. rGIN (Sato et al., 2020) pairs each node with a random feature, thereby breaks locally indistinguishable subgraphs. Our method is different from previous approaches in that we exploit the information of entire graphs' node features, achievable with a minimal network depth of 2.

Memory-based neural networks date back to the 1990s (for example (Schmidhuber, 1992)). Notable recent developments include Neural Turing Machine (Graves et al., 2014) and Memory Networks (Weston et al., 2014). Our memory augmentation mechanism could be viewed as a GNN implementation of the design principle in (Weston et al., 2014). There has been a couple of works on GNNs that adopted memory-based design lately (Khasahmadi et al., 2020; Ma et al., 2019a), under different problem contexts.

## 5    EXPERIMENTS

In this section we evaluate the performance of memory augmented GNNs using two canonical architectures MemGAT and MemGCN. We also conduct an ablation study to decompose performance gains under different design aspects. We use four types of benchmark datasets, with their characteristic statistics summarized in appendix A.1:

**Citation networks** Cora, Citeseer, and Pubmed are standard citation network benchmark datasets. For all the three datasets, graphs are constructed by treating documents as nodes and citation links as edges, the task is to classify academic papers into different subjects using the papers' bag-of-words representation as features

**Actor co-occurrence networks** This dataset is the actor-only induced subgraph of the film-director-actor-writer network. Where nodes correspond to actors, and an edge between two nodes denotes co-occurrence on the same Wikipedia page. Node features correspond to some keywords in the Wikipedia pages and labels are actors categories

**Web networks** We use Cornell, Texas, and Wisconsin dataset (Pei et al., 2020). where nodes and edges represent web pages and hyperlinks, respectively. The feature of each node is the bag-of-words representation of the corresponding page, and labels are categories of web pages.

**Protein-protein interaction (PPI)** This is a dataset containing 24 graphs, with each graph corresponding to a human tissue. (Hamilton et al., 2017). The task is to classify protein functions based on features containing positional gene sets, motif gene sets and immunological signatures. This setting is inductive in that at test time, algorithms shall make prediction with respect to completely unseen graphs. We will present the details of the PPI experiment in appendix A.6.

### 5.1 EXPERIMENTAL SETUP

**Architecture and parameters** We mostly adopt parameters for backbone architectures from their original implementation, respectively GAT (Veličković et al., 2018) and GCN (Kipf & Welling, 2016), detailed in appendix A.1. We applied entropy regularization (Grandvalet & Bengio, 2005) with tuning parameter $0.6$ for Cora and Citeseer datasets and $0.1$ for Pubmed dataset. We use the following truncated diffusion matrix (Klicpera et al., 2019b) to generate edge weights:

$$S := S(\theta, K, T) := \sum_{k=0}^{K} \theta_k T^k \tag{4}$$

In our experiments we use two sets of $\theta$s: *personalized page rank*: $\theta_k = \alpha(1-\alpha)^k$ with $\alpha = 0.15$ and *heat kernel*: $\theta_k = \frac{t^k e^{-t}}{k!}$ with $t = 5$. We choose the transition matrix $T$ to be either $A_{\text{rw}} = (D+I)^{-1}(A+I)$ or $A_{\text{sym}} = (D+I)^{-1/2}(A+I)(D+I)^{-1/2}$. We report the best performing combination for each experiment. We tune the number of random walk steps $K \in \mathbb{N}$. Note that for large $K$, the resulting matrix is an approximation to the diffusion matrix (Klicpera et al., 2019b).

For the memory component, we used $\lambda = 0.9$, we choose the number of memory nodes to be the number of classes across all tasks, a study of this hyperparameter is presented in appendix A.3. Finally, we applied skip connection using the form proposed in Zhang & Meng (2019) to stabilize training. The detailed tuning strategies are listed in appendix A.1. For a better understanding of performance gains corresponding to different modifications to the backbone architecture, we provide an ablation study on the citation datasets. For the rest of the datasets, we additionally report performances without using edge weights, which we termed MemGAT-NE and MemGCN-NE.

**Baseline comparisons** We compare MemGAT and MemGCN with their backbone architectures across all benchmarks. We also report two state-of-the-art models that are able to aggregate neighborhood information from distant nodes: APPNP (Klicpera et al., 2019a) and GCNII (Chen et al., 2020). For the citation networks, we additionally report GraphSage (Hamilton et al., 2017) with max-pooling aggregation and GIN (Xu et al., 2019) and G$^3$NN (Ma et al., 2019b). For actor and wiki networks, we additionally report GEOM-GCN (Pei et al., 2020) with its best performing variant. Besides, we report the performance of a two-layer MLP with 256 hidden units on these datasets, as was recently noted that MLP performs very well on actor and wiki networks (Zhu et al., 2020).

**Evaluation strategy** it was reported in (Shchur et al., 2018) that the performance of current state-of-the-art graph neural networks are typically unstable with respect to different train/validation/test split of the dataset. Hence for fair comparison of algorithm performances, in addition to results on the original, standard split, we provide extensive results over 10 random splits of Cora, Citeseer and Pubmed datasets.

### 5.2 RESULTS

We use accuracy for performance metrics across all the tasks. The results of our evaluation experiments are summarized in Tables 1 and 2. We report training details in appendix A.1. As shown in

Table 1: Summary of results (%mean $\pm$ %standard deviation test set accuracy) for the citation datasets, with 10 random train/validation/test split of datasets as well as the standard split. We annotate the original paper where we take the results from aside the algorithm name

| Algorithm | Cora | Citeseer | Pubmed |
|---|---|---|---|
| **Random split** | | | |
| GCN (Shchur et al., 2018) | $81.5 \pm 1.3$ | $71.9 \pm 1.9$ | $77.8 \pm 2.9$ |
| GraphSage(maxpool) (Shchur et al., 2018) | $76.6 \pm 1.9$ | $67.5 \pm 2.3$ | $76.1 \pm 2.3$ |
| GAT (Shchur et al., 2018) | $81.8 \pm 1.3$ | $71.4 \pm 1.9$ | $78.7 \pm 2.3$ |
| GIN (Klicpera et al., 2019b) | $73.96 \pm 0.46$ | $61.09 \pm 0.58$ | $72.38 \pm 0.63$ |
| APPNP (our run) | $82.45 \pm 1.89$ | $70.60 \pm 1.31$ | $77.28 \pm 3.14$ |
| MemGAT(ours) | $\mathbf{83.94 \pm 1.04}$ | $\mathbf{74.07 \pm 1.1}$ | $\mathbf{79.18 \pm 2.11}$ |
| MemGCN(ours) | $82.94 \pm 1.61$ | $72.87 \pm 1.69$ | $78.53 \pm 1.91$ |
| **Standard split** | | | |
| GCN (Shchur et al., 2018) | $81.9 \pm 0.8$ | $69.5 \pm 0.9$ | $79.0 \pm 0.5$ |
| GraphSage(maxpool) (Shchur et al., 2018) | $77.4 \pm 1.0$ | $67.0 \pm 1.0$ | $76.6 \pm 0.8$ |
| GAT (Shchur et al., 2018) | $82.8 \pm 0.5$ | $71.0 \pm 0.6$ | $77.0 \pm 0.3$ |
| APPNP (our run) | $83.60 \pm 0.61$ | $72.46 \pm 0.52$ | $79.04 \pm 0.53$ |
| G$^3$NN(GAT) (Ma et al., 2019b). | $82.9 \pm 0.3$ | $74.0 \pm 0.3$ | $77.4 \pm 0.4$ |
| G$^3$NN(GCN) (Ma et al., 2019b). | $82.2 \pm 0.3$ | $74.5 \pm 0.3$ | $78.4 \pm 0.4$ |
| GCNII (Chen et al., 2020) | $\mathbf{85.5}$ | $73.4$ | $80.3$ |
| MemGAT(ours) | $84.65 \pm 0.52$ | $74.20 \pm 0.73$ | $79.18 \pm 0.56$ |
| MemGCN(ours) | $84.30 \pm 0.53$ | $\mathbf{75.12 \pm 0.22}$ | $\mathbf{80.46 \pm 0.27}$ |

Table 2: Summary of results (%mean $\pm$ %standard deviation test set accuracy) for actor co-occurence and web network datasets, with 10 random train/validation/test split of datasets. We annotate the original paper where we take the results from aside the algorithm name

| Algorithm | Actor | Cornell | Texas | Wisconsin |
|---|---|---|---|---|
| GCN (Pei et al., 2020) | $26.86$ | $52.70$ | $52.16$ | $45.88$ |
| GAT (Pei et al., 2020) | $28.45$ | $54.32$ | $58.38$ | $49.41$ |
| APPNP (Chen et al., 2020) | $31.31$ | $73.51$ | $65.41$ | $69.02$ |
| GEOM-GCN (Pei et al., 2020) | $31.63$ | $60.81$ | $67.57$ | $64.12$ |
| GCNII (Chen et al., 2020) | NA | $76.49$ | $77.84$ | $81.57$ |
| MLP (our run) | $\mathbf{37.01 \pm 1.02}$ | $80.65 \pm 6.34$ | $80.84 \pm 4.21$ | $84.31 \pm 3.41$ |
| MemGAT(ours) | $31.93 \pm 1.69$ | $70.37 \pm 7.57$ | $72.08 \pm 8.02$ | $73.80 \pm 5.05$ |
| MemGCN(ours) | $36.95 \pm 0.95$ | $\mathbf{81.92 \pm 6.00}$ | $82.19 \pm 5.19$ | $\mathbf{84.96 \pm 3.59}$ |
| MemGAT-NE(ours) | $34.93 \pm 0.74$ | $66.14 \pm 2.99$ | $61.75 \pm 3.18$ | $65.14 \pm 5.41$ |
| MemGCN-NE(ours) | $36.44 \pm 0.53$ | $80.76 \pm 0.99$ | $\mathbf{82.76 \pm 2.14}$ | $83.35 \pm 1.12$ |

Table 3: Summary of the ablation study results (%mean $\pm$ %standard deviation test set accuracy) for the citation datasets over 100 trials. For MemGCN, we did not use skip connection on the citation datasets; for MemGAT, we did not incorporate edge weights on the pubmed dataset.

| Model | MemGAT | | | MemGCN | | |
|---|---|---|---|---|---|---|
| **Dataset** | Cora | Citeseer | Pubmed | Cora | Citeseer | Pubmed |
| (I) | $82.32 \pm 0.57$ | $71.11 \pm 0.97$ | $77.00 \pm 0.30$ | $81.99 \pm 0.52$ | $69.50 \pm 0.90$ | $79.00 \pm 0.50$ |
| (II) | $83.54 \pm 0.55$ | $71.76 \pm 0.48$ | N/A | $82.26 \pm 0.88$ | $72.68 \pm 0.27$ | $79.95 \pm 0.23$ |
| (III) | $83.77 \pm 0.72$ | $73.21 \pm 0.66$ | $78.97 \pm 1.05$ | N/A | N/A | N/A |
| (IV) | $84.65 \pm 0.52$ | $74.20 \pm 0.73$ | $79.18 \pm 0.56$ | $84.30 \pm 0.53$ | $75.12 \pm 0.22$ | $80.46 \pm 0.27$ |

table 1, MemGAT and MemGCN consistently outperforms GAT and GCN by a significant margin and is competitive to current state-of-the-art algorithms for graph learning. In particular, on the web network datasets, MemGCN improves GCN with a relative accuracy gain of over 50% while using the same network depth, and outperforms GCNII, which uses deep architectures (16 and 32 layers) on these datasets. Moreover the gain of using edge weights appears marginal on the web datasets, suggesting that the incorporation of memory component is highly beneficial. Results from random split experiments demonstrate that memory augmented architectures have fairly stable performance.

**Ablation study** We analyze the contribution of different components of MemGAT or MemGCN in a progressive way, with (I) backbone GNN (original GAT or GCN) (II) backbone GNN with edge weights. (III) backbone GNN with edge weights and skip connections. (IV) full MemGAT or MemGCN, which corresponds to adding memory component to the setup in (III). Table 3 records results on the citation datasets. The results demonstrate significant improvement provided by incorporating the memory component.

## 6 CONCLUSIONS

We introduced memory augmentation, a framework that extends GNNs with a memory component that aims at incorporating global information of the graph. Our method has the advantage of stronger expressive power than GNNs based on LUMP protocol and is applicable to standard GNN architectures like GAT and GCN. Experimental results reflect our theoretical motivations.

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

## A  Appendix

### A.1  Experimental details

**Dataset summary**     An overview of the characteristics of the datasets we used for our experiments is given in Table 4.

Table 4: Summary of the datasets used in our experiments.

|  | Cora | Citeseer | Pubmed | Actor | Cornell | Texas | Wisconsin |
|---|---|---|---|---|---|---|---|
| # **Nodes**(# **Graphs**) | 2708 (1) | 3327 (1) | 19717 (1) | 7600 (1) | 183 (1) | 183 (1) | 251 (1) |
| # **Edges** | 5429 | 4732 | 44338 | 33544 | 295 | 309 | 499 |
| # **Features/Node** | 1433 | 3703 | 500 | 931 | 1703 | 1703 | 1703 |
| # **Classes** | 7 | 6 | 3 | 5 | 5 | 5 | 5 |
| # **Training Nodes** | 140 | 120 | 60 | 60% | 60% | 60% | 60% |
| # **Validation Nodes** | 500 | 500 | 500 | 20% | 20% | 20% | 20% |
| # **Test Nodes** | 1000 | 1000 | 1000 | 20% | 20% | 20% | 20% |

For Actor, Cornell, Texas and Wisconsin datasets, we used the available split given in https://github.com/graphdml-uiuc-jlu/geom-gcn/tree/master/splits to ensure fair comparison.

**Backbone architecture**     Here we specify setups of the backbone architectures.

**MemGAT**     We mostly adopted the original architecture of GAT. Specifically, we used a two-layer GAT across all tasks. For all but the PubMed dataset, the first layer consists of 2 attention heads and 256 features each. The second layer is used for classification with a single attention head that computes $C$ features (with $C$ being number of classes), followed by a softmax activation. For Pubmed dataset, we used 8 output attention heads as in Veličković et al. (2018). For the edge weights, we picked $\psi_{ij} = S_{ij}$ where $S$ is the edge weight matrix, and $\phi^{(l)}(h, h') = \mathsf{LeakyRelu}\left(S_{ij}\langle a, \left[W^{(l)}h \parallel W^{(l)}h'\right]\rangle\right)$. The nonlinear function $\sigma^{(l)}$ was chosen identically across all models and all layers as the ELU function Clevert et al. (2015), with the exception that we did not use nonlinear transform of the input features.

**MemGCN**     We used a two-layer GCN across all tasks with 256 hidden features.

**Training**     We applied $L_2$ regularization with tuning parameter $0.001$ and a dropout Srivastava et al. (2014) operation of probability $0.6$ to each layer's inputs for all transductive tasks. We applied entropy regularization Grandvalet & Bengio (2005) with tuning parameter $0.6$ for Cora and Citeseer datasets (but not for the ablation study) and $0.1$ for Pubmed dataset. All models used Glorot initialization Glorot & Bengio (2010) and cross entropy loss optimized using Adam Kingma & Ba (2014) with an initial learning rate of $0.01$ for Pubmed dataset, and $0.005$ for all the other datasets. We used an early stopping strategy on both the cross-entropy loss and accuracy on the validation nodes, with a patience of 100 epochs.

**Optimal hyperparameters**   We report the optimal choice of hyperparameters in table 5.

Table 5: Optimal choice of hyperparameters, namely the incorporation of skip connection, the number of random walk steps $K$, and the type of transition matrix $T$. We tune $K$ over the set $\{0, 1, 2, 3, 4, 5, 10, 20, 30\}$.

| | **MemGAT** | | | **MemGCN** | | |
|---|---|---|---|---|---|---|
| **Dataset** | Cora | Citeseer | Pubmed | Cora | Citeseer | Pubmed |
| skip connection | ✓ | ✓ | ✓ | ✗ | ✗ | ✗ |
| $K$ | 30 | 30 | 0 | 3 | 1 | 30 |
| $T$ | $A_{\text{sym}}$ | $A_{\text{sym}}$ | $A_{\text{sym}}$ | $A_{\text{sym}}$ | $A_{\text{sym}}$ | $A_{\text{sym}}$ |

| | **MemGAT** | | | | **MemGCN** | | | |
|---|---|---|---|---|---|---|---|---|
| **Dataset** | **Actor** | **Cornell** | **Texas** | **Wisconsin** | **Actor** | **Cornell** | **Texas** | **Wisconsin** |
| skip connection | ✓ | ✓ | ✓ | ✓ | ✓ | ✓ | ✓ | ✓ |
| $K$ | 0 | 3 | 3 | 3 | 3 | 3 | 0 | 3 |
| $T$ | $A_{\text{sym}}$ | $A_{\text{sym}}$ | $A_{\text{sym}}$ | $A_{\text{sym}}$ | $A_{\text{sym}}$ | $A_{\text{sym}}$ | $A_{\text{sym}}$ | $A_{\text{sym}}$ |

**Implementation**   We implemented MemGAT based on the open source PyTorch Paszke et al. (2019) implementation of GAT Veličković et al. (2018) at `https://github.com/PetarV-/GAT`, and MemGCN based on the open source PyTorch implementation of GCN (Kipf & Welling, 2016) at `https://github.com/tkipf/pygcn`,

## A.2   PROOF OF THEOREMS

*Proof of theorem 1.* We show by induction on $l$, for $l = 0$ it follows trivially since $X_v = X'_{f(v)}, \forall v \in V$, suppose for $l = L$ we have $h_v^{(L)} = h_{f(v)}^{(L)\prime}, \forall v \in V$, for $l = L + 1$, consider any $v \in V$, since $\mathsf{STAR}(v)$ is isomorphic to $\mathsf{STAR}(f(v))$ and the map $f$ is surjective, it follows that the multiset representation of the feature vector $X_{\mathcal{N}_v}$ is identical to $X'_{\mathcal{N}_{f(v)}}$, thus an unordered aggregation function would produce $\tilde{h}_v^{(L)} = \tilde{h}_{f(v)}^{(L)\prime}$, we conclude that $h_v^{(L+1)} = \mathsf{COMBINE}\left(h_v^{(L)}, \tilde{h}_v^{(L)}\right) = \mathsf{COMBINE}\left(h_{f(v)}^{(L)\prime}, \tilde{h}_{f(v)}^{(L)\prime}\right) = h_{f(v)}^{(L+1)\prime}$. □

*Proof of lemma 1.* For part (i), under condition $\mathbf{C}_1$ there exists a parameter (hereafter referred to as identifier) $\vartheta_0^* = \Theta(\mathcal{X})$ that identifies every bounded subset (with subset in the sense of multiset) of $\mathcal{X}$ up to distributional equivalence. The map $\Theta$ hence maps feature set to an "identifier" parameter. Since $\mathcal{X}$ is a countable subset of some euclidean space, it's easy to find an element $\mathbf{m_0} \notin \mathcal{X}$, and we let $\bar{\mathcal{X}} = \mathcal{X} \bigcup \{\mathbf{m_0}\}$, it follows immediately that if we augment every $\{X\}$ into $\{X \bigcup \{\mathbf{m_0}\}\}$, the identifier $\tilde{\vartheta}_0^* = \Theta(\bar{\mathcal{X}})$ over $\bar{\mathcal{X}}$ identifies $\{X \bigcup \{\mathbf{m_0}\}\}$ and $\{X' \bigcup \{\mathbf{m_0}\}\}$ for any $\{X\} \neq \{X'\}$, since $\mathbf{m_0}$ always has a multiplicity of one and is distinct from all elements in $\mathcal{X}$.

The injectivity is therefore defined in the following sense: for any multiset $\left\{\tilde{X}\right\}$ satisfying:

(i) Its underlying set $\tilde{X}$ represented as $\tilde{X} = \{\mathbf{m_0}\} \bigcup X$ where $X$ is a subset of $\mathcal{X}$ with bounded size.

(ii) The multiplicity of $\mathbf{m_0}$ is restricted to be one, and the multiplicities of other elements are uniformly bounded from above.

Then under identifier $\tilde{\vartheta}_0^*$, $\mathsf{GNN}_{\tilde{\vartheta}_0^*}\left(\left\{\tilde{X}\right\}\right) = \mathsf{GNN}_{\tilde{\vartheta}_0^*}\left(\left\{\tilde{X}'\right\}\right)$ if and only if $\left\{\tilde{X}\right\} = \left\{\tilde{X}'\right\}$, which is equivalent to $\{X\} = \{X'\}$. Applying the previous argument iteratively, we obtain identifier for each layer $\tilde{\vartheta}_l^*, l \in \mathbb{N}$ and injectivity could be defined in similar ways. Part (ii) is a consequence of part (i) in that we choose $X$ and $X'$ to be the corresponding graph feature of $G$ and $G'$. □

*Proof of theorem 2.* By lemma 1, the output of the second MemGAT layer would be different for $h_v^{(2)}, h_{f(v)}^{(2)\prime}, \forall v \in V$, since the underlying feature space is countable, there exists a nonlinear function $\sigma$ satisfying $\sigma\left(h_v^{(2)}\right) < 0.5, \sigma\left(h_{f(v)}^{(2)\prime}\right) \geq 0.5, \forall v \in V$. Picking the readout function as $\sigma$ finishes the proof. Note also that this function could be approximated by universal approximators like multi layer perceptrons. □

*Proof of corollary 1.* We first show for GCN. Note that GCN has several different definitions, we will follow the general definition in Dehmamy et al. (2019) without bias term:

$$H^{(l+1)} = \sigma\left(\tau(A)H^{(l)}W\right) \tag{5}$$

where $H^{(l)}$ is the matrix stacked by hidden representations of each node in the $l$th layer, and $\tau : \mathbb{R}^{N \times N} \mapsto \mathbb{R}^{N \times N}$ is a matrix transformation operation. With $D = \mathrm{diag}(AI_N)$, two popular forms of GCN are defined as

**Kipf & Welling (2016)** uses $\tau_1(A) = (D + I_n)^{-1/2}(A + I_N)(D + I_N)^{-1/2}$, and $\sigma$ is RELU.

**Xu et al. (2019)** uses $\tau_2(A) = (D + I_n)^{-1}(A + I_N)$ which reduces to mean pooling, and $\sigma$ is RELU.

The fact that GCN formulated by $\tau_2$ satisfied condition $\mathbf{C}_1$ is directly implied by Xu et al. (2019, Corollary 8). But for GCN induced by $\tau_1$, the identifiability result need not hold since the aggregation process of each node $v \in V$ is determined not solely by its neighborhood information, but also by the degree of its neighborhoods which could be arbitrary. Nevertheless, we could still gain insights from this (more popular) design by noting that with respect to regular graphs, the identifiability issue of both formulations are the same, and are mitigated via memory augmentation.
For GAT, consider the worst case of two multisets with their underlying set identical with a single element but different in multiplicities. In this case, regardless of the attention mechanism, GAT is identical to mean pooling. Hence it suffices to choose the identifier obtained from (Xu et al., 2019, Corollary 8) over mean pooling that makes GAT identify multisets up to distributional equivalence. □

### A.3 ON THE EFFECT OF NUMBER OF MEMORY NODES

In this section we present a study on training MemGAT on the Cora dataset using different number of memory nodes $M$. The tuning range is $\{1, 3, 5, 7, 9, 11, 13, 15, 17\}$. The rest of the hyperparameters are the same as the one reported in table 1. The results are reported in table 6. The result shows little performance difference in using different number of memory nodes.

### A.4 ARCHITECTURES UNSUITABLE FOR MEMORY AUGMENTATION

GNN architectures that utilize max pooling for aggregating operation may not identify distributionally equivalent instances (Xu et al., 2019, Corollary 9), hence the max-pooling version of GraphSAGE (Hamilton et al., 2017) is not a proper backbone architecture for memory augmentation. GIN (Xu et al., 2019) uses sum pooling that is strictly more expressive than mean pooling hence satisfies condition $\mathbf{C}_1$. However, summing up feature vectors of the whole graph increases numerical instability and is empirically found hard to train.

### A.5 COMPARISONS WITH OTHER MESSAGE PASSING VARIANTS

**Comparison with CPCGNN** CPCGNN Sato et al. (2019) utilizes a *consistent port numbering* that numbers the neighbors of each node $v$ by an integer $i \in [\mathsf{degree}(v)]$, according to a *port numbering function* $p$ such that $p(v, i) = (u, j)$ identifies the neighboring node $u$ labeled $i$ and a port number $j \in [\mathsf{degree}(u)$. The port numbering rule is said to be *consistent* if $p(p(v, i)) = (v, i)$ for any valid $(v, i)$ pairs. CPCGNN allows node $v$ sending messages to node $u$ depending on both its own feature and the port number of $u$, thus forms a certain kind of locally ordered message passing framework that is strictly more expressive than locally unordered GNNs. However was shown in

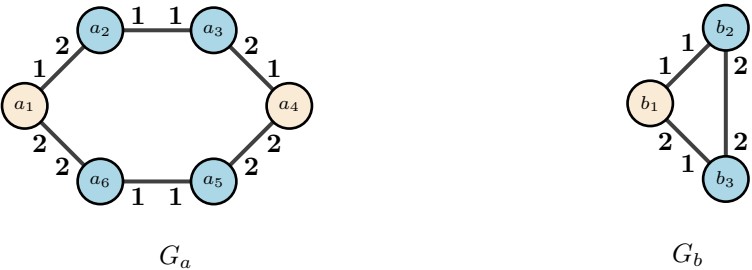

Figure 4: Consistent port numberings for $G_a$ and $G_b$ that makes them locally distinguishable

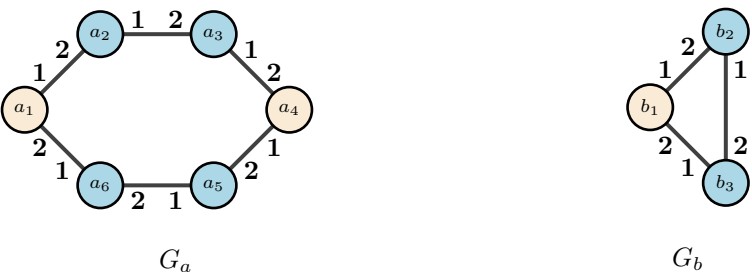

Figure 5: Consistent port numbering for $G_a$ and $G_b$ that fails to distinguish $a_1, a_4$ and $b_1$

Garg et al. (2020) that since consistent port numbering functions are *non-unique*, there exists some port numbering functions that does not strengthen expressiveness, we illustrate this phenomenon using the construction in figure 4 and figure 5. Figure 4 shows a port numbering that makes $a_1, a_4$ receiving different messages with that of $b_1$. Meanwhile in figure 5 the port numbering can not distinguish $a_1, a_4$ and $b_1$. Finding a consistent port numbering that succeeds in distinguishing local structures is yet another challenging task, MemGAT thus offers an easier choice when the two graphs have different global features.

**Comparison with DimeNet** DimeNet Klicpera et al. (2020) is a *directional* message passing model that exploits the relative layout of local neighborhood through angles. Specifically DimeNet computes node embedding $h_v^{(l)}$ as the summation of its incoming message embeddings $h_v^{(l)} = \sum_{u \in \mathcal{N}_v} m_{uv}^{(l)}$, and the update rule is defined as

$$m_{uv}^{(l)} = f_{\text{update}}\left(m_{uv}^{(l-1)}, \sum_{w \in \mathcal{N}_v \setminus u} f_{\text{integrate}}\left(m_{wv}^{(l-1)}, e^{(uv)}, a^{(wu,uv)}\right)\right) \quad (6)$$

where $f_{\text{integrate}}$ and $f_{\text{update}}$ are analogs of aggregate and combine as in LUMP protocol, $e^{(uv)}$ is a representation vector measuring the distance from $u$ to $v$, and $a^{(wu,uv)}$ combines $\angle wuv$ with the distance from $w$ to $u$. The choice of metric is problem dependent, and we presume a suitable one exists. Consider the following construction:

Table 6: Study on number of memory nodes on the Cora dataset using MemGAT model. Results (%mean ± %standard deviation test set accuracy) are computed over 100 trials

| $M$ | Performance |
|---|---|
| 1 | $84.60 \pm 0.58$ |
| 3 | $84.62 \pm 0.58$ |
| 5 | $84.70 \pm 0.52$ |
| 7 | $84.64 \pm 0.52$ |
| 9 | $84.71 \pm 0.54$ |
| 11 | $84.69 \pm 0.67$ |
| 13 | $84.78 \pm 0.59$ |
| 15 | $84.78 \pm 0.59$ |
| 17 | $84.76 \pm 0.60$ |

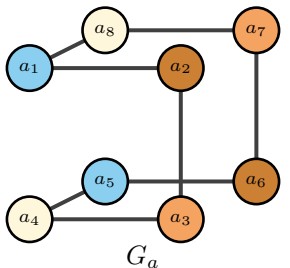 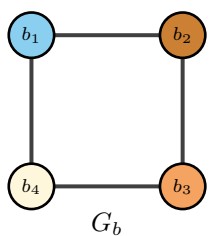

Figure 6: Two graphs $G_a = (V_a, E_a)$ and $G_b = (V_b, E_b)$ that DimeNet cannot distinguish locally: for any $V \in \{V_a, V_b\}$ and all $w, u, v \in V$ satisfying $(u, w) \in E, (u, v) \in E$, the graph is constructed such that $\angle wuv = \pi/2$ and for any $(u, v) \in E$, the distance between $u$ and $v$ is identical.

Figure 6 shows a construction that DimeNet fails to distinguish with the local isomorphism map defined as $f(a_1) = f(a_5) = b_1, f(a_2) = f(a_6) = b_2, f(a_3) = f(a_7) = b_3, f(a_4) = f(a_8) = b_4$, while MemGAT is able to distinguish them.

The above contrived examples suggest that the optimal choice of GNN architecture shall be problem dependent.

### A.6 EXPERIMENT ON THE PPI DATASET

#### A.6.1 EXPERIMENTAL SETUP

We evaluated the results of MemGAT and MemGCN in the PPI dataset. For MemGAT, we used a three-layer network architecture, the first two layers consist of 4 attention heads computing 256 features, and a final layer uses 4 attention heads computing 121 features each. For MemGCN, we used a two-layer network with 256 hidden features. Since the construction process of PPI dataset already includes diffusion like mechanisms, and was previously reported to have no improvements Klicpera et al. (2019b), we did not apply diffusion in this experiment. The training setup is identical to those used in transductive tasks.

**Baseline comparisons** Aside from two backbone architectures, we report GCN/GAT version of JK-Net Xu et al. (2018) with the best result (LSTM aggregation) and GeniePath Liu et al. (2019) that utilizes attention style design in the breadth search phase.

#### A.6.2 RESULTS

We present results on the PPI dataset on table 7. The result provides strong evidence that incorporating global graph information significantly improves node classification. Moreover, since diffusion is not used, the captured local structure is the same with that of backbone architectures, and smaller than the other variants. Therefore we think the contribution of memory mechanism is significant in this case, suggesting evidence that the global information helps for inductive node classification.

Table 7: Summary of the results (% test set micro-averaged $F_1$ score) for the inductive setting, performance metric is averaged over 100 trials, with standard deviations also reported

| Algorithm | PPI |
|---|:---:|
| GCN Hamilton et al. (2017) | 50.00 |
| GraphSage(LSTM) Veličković et al. (2018) | 76.8 |
| GAT Veličković et al. (2018) | $96.80 \pm 0.20$ |
| JK-LSTM(GCN)Xu et al. (2018) | 81.8 |
| JK-LSTM(GAT)Xu et al. (2018) | $97.60 \pm 0.7$ |
| GeniePathLiu et al. (2019) | 97.9 |
| MemGAT(ours) | $\mathbf{98.47 \pm 0.15}$ |
| MemGCN(ours) | $88.01 \pm 0.40$ |

