# OpenReview forum: "Memory Augmented Design of Graph Neural Networks"
_ICLR.cc/2021/Conference — Reject_

### Official Review · AnonReviewer2 · 2020-10-26
**This paper addresses a very special case that rarely happens in practice**

**Rating:** 5
**Confidence:** 4

**Review:**

This paper proposes to enhance GNNs with memory networks. The authors prove that existing GNNs cannot distinguish two nodes in two graphs which are local isomorphic, then they propose introducing additional nodes for a graph that connect to all existing nodes, which help distinguish neighborhood structure for the given two nodes. This paper is basically well written and easy to follow. The idea of combining memory networks and GNNs is also novel and interesting. The experimental results demonstrate the efficacy of the proposed method. However, I have the following concerns:

1. How should we understand local isomorphism intuitively? What are the connection and difference between the conventional isomorphism and local isomorphism? Can the proposed analysis be generalized to the conventional graph isomorphism? I think the paper goes a little quick from Definition 1 to Theorem 1. It would be better to put more words here to clearly demonstrate the concept of local isomorphism.

2. The authors provide an example in Figure 1 and 2 to show the limit of existing GNNs and that an additional memory node is able to distinguish the two subtrees. However, the example only works in the case of two graphs. In a single graph, adding an additional node that connects to all nodes does not provide such ability to distinguish two nodes. Since the used datasets are all single graph, I was wondering what the benefit of introducing an additional node is.

3. The theoretical part is nice to read, however, they are all existence theorems, which neither tells us how to find such solutions or provides performance guarantees for the proposed method. The proposed method seems more likely to address a very special case where two nodes (in two graphs respectively) cannot be distinguished, but this case rarely happens in practice.

4. In experiments, I was wondering why the authors uses diffusion matrix rather than the original adjacency matrix, since it makes readers unclear that the performance gain is brought by the memory mechanism or the diffusion matrix. It would be more convincing to use original adjacency matrix in the proposed method, or add (Klicpera et al., 2019) as a baseline.

---

> ### Author Response · Authors · 2020-11-22
> **Response**
>
> We thank the reviewer for providing comments. Below we address your specific points:
>
> (#1) Interpreting local isomorphism: local isomorphism is a relaxed notion of graph isomorphism: if two graphs are isomorphic, the map that characterizes the isomorphism relation also characterizes the local isomorphism, but the converse is not true as seen in figure 1. Since the primal task, we studied in the paper is node classification, which reflects more "local" considerations than graph-level tasks like graph classification (for which the graph isomorphism problem is used for measuring expressivity), using local isomorphism is a useful notion in characterizing predictions that are local. More specifically, to characterize expressive power under classification tasks a commonly adopted framework is to assess if the proposed function class is able to express all allocations of class labels to the training dataset, following this intuition, it makes sense to consider under which circumstances will two different allocations never get discriminated. To this end, locally isomorphic graphs give a solid example------the predictions given by any LUMP GNN over locally isomorphic graphs are everywhere identical w.r.t. the local isomorphism map, while the two graphs could be not isomorphic to each other.
>
> (#2) Multi-graph evidence: we provide evaluations on the PPI dataset in the revision. The result suggests significant improvement to the backbone architectures by incorporating memory components.
>
> (#3) The practicality of locally indistinguishable graphs: we refer to remark 1 in the revised version of the paper for details.
>
> (#4) On the usage of the diffusion matrix: For the citation networks, we provided an ablation study that characterizes the additional performance gain of the incorporation of memory components. For the rest datasets, we've additionally provided the results using the original adjacency matrix in the revision. The results suggest solid improvements over backbone architectures

---

### Official Review · AnonReviewer4 · 2020-10-28
**Review comments on Paper 2966**

**Rating:** 5
**Confidence:** 4

**Review:**

==========Summary==========

In this paper, the authors study how to enhance the expressive power of GNNs by memory augmentation. In particular, the authors focus on the cases of the "locally indistinguishable" property, demonstrate why existing GNNs fail to differentiate such structures, and propose a memory augmentation strategy to break the tie. Starting from theoretical analysis, the authors suggest how to apply memory augmentation to GCN and GAT. Empirical results on public benchmark datasets suggest the effectiveness of the proposed method.

==========Reason for the rating==========

At this moment, I am standing between 5 and 6. Overall, the authors tackle the expressive power enhancement problem from an interesting angle, and the technical insights are well presented. My main concern is on the gap between real-life data distribution and theoretical assumptions. While it is theoretically interesting, it is unclear how much difference "locally indistinguishable" structures make in reality.

==========Strong points==========

1. The authors investigate how to enhance the expressive power of GNNs from a unique perspective.

2. The core idea is well presented with logical reasoning based on theoretical and concrete evidences.

3. Empirical results suggest the proposed technique is promising in node classification tasks.

==========Weak points==========

1. The authors may need to clarify the scope of the theoretical results. Do the theoretical result still hold for the cases where node attributes are multi-dimensional numerical values? For the assumption in Section 3.1 "We will assume the set $\mathcal{X}$ to be countable, then by (Xu et al., 2019, Lemma 4), the range of $h^{(l)}_v$ , denoted $\mathcal{H}^{(l)}$, is also
countable for any given parameterization and any layer l.", what is the exact implication?

2. The authors may share a few concrete examples from real-life data that suggest it is valuable to differentiate locally indistinguishable structures. Although the prediction results suggest the effectiveness, it is critical to confirm it is locally indistinguishable structures that make this difference.

3. For the results presented in Table 1 and Table 2, the authors may need to consider identical train/validation/test splits for all the baseline methods for fair comparison.

==========Questions during rebuttal period==========

Please address and clarify the weak points above.

==========Post rebuttal==========

I appreciate the authors' effort on answering my questions. Meanwhile, the authors' response does not fully address my concerns. I keep my rating as it is.

---

> ### Author Response · Authors · 2020-11-22
> **Response**
>
> We thank the reviewer for providing comments. Below we address your specific points:
>
> (Weak points #1 ) On the necessity of the countable feature space assumption: the methodology of assuming a countable feature universe was suggested in Zaheer et al, 2017 and Xu et al, 2019. Using this assumption would make the characterization of expressivity much easier as one is able to construct explicitly an identification function (see Lemma 5 in Xu et al, 2019 or Theorem 2 in Zaheer et al, 2017). The extension to the uncountable cases is highly non-trivial as discussed in section A.2 in Zaheer et al, 2017, and the discussion carries over to multisets. Note that for features that represent counts or bag-of-words like features, the countable universe assumption is appropriate.
>
> (Weak points #2 ) The practicality of locally indistinguishable graphs: we refer to remark 1 in the revised version of the paper for details. To precisely characterize the power of memory-augmented designs over datasets that contains many locally indistinguishable structures, we need to conduct additional experiments over datasets from biochemical domains. Due to time limits, we did not include the study in the current version of the paper.
>
> (Weak points #3 ) On the reproducibility of random splits: for splits in table 2, the authors of GEOM-GCN provided pre-computed splits https://github.com/graphdml-uiuc-jlu/geom-gcn/tree/master/splits, for results in table 1 we copied most of the random split results from cross-reference. Note that the random split setup corresponds to perform 10 random splits with 100 replications each (altogether 1000 replications for a single dataset). The result is fairly stable with respect to variations in both split choice and training randomness so we didn't reproduce all the baseline independently.
>
> References:
> [1] Zaheer et al, Deep sets, NIPS, 2017
> [2] Xu et al, How powerful are graph neural networks?, ICLR, 2019

---

### Official Review · AnonReviewer1 · 2020-10-29
**Interesting model augmentations for GNNs**

**Rating:** 5
**Confidence:** 4

**Review:**


The paper proposes a new class of neural networks augmented by memory nodes. The authors studied variants of GCN, and GAT with memory nodes and specifically designed edge attention functions and aggregators. The proposed models achieved competitive performance compared to state-of-the-art methods.

Clarity.
Overall, the paper is well-written. Theoretical results are introduced with concise discussions and pictorial examples. They made the results more accessible.

Quality/significance (pros)
	1. The authors found the cases where LUMP-GNNs fail and they prove it in theorem 1.
	2. The authors theoretically and empirically prove that memory-augmentation helps to identify locally indistinguishable graphs with different multiset feature representations in theorem 2 and Figure 3.
	3. The ablation study shows that the proposed modules are effective and provide significant improvement in some settings.

Weaknesses (cons)
	1. The authors should compare their work with GNNs with non-local operations, e.g., LatentGNN [1]. The paper also studies the limitations of local GNNs (not specifically LUMP) but the resulting model is similar to memory augmented GNNs and it has skip connections and augmented by convolution in the latent node space.
	2. It's an interesting technique to improve the expressive power of GNNs but the augmentation requires significant modifications of the original GNNs. If the technique is applicable to general GNNs in a plug-and-play manner, it would be more useful.
	3. Depending on the edge weights, the models may behave differently. The handcrafted edge weights from the truncated diffusion matrix naturally raise the question of whether they are necessary to show the effectiveness of the proposed technique.

Question.
	1.  In the aggregation scheme of MemGCN, $M_v^{(l=1)} = \lambda W^{(l)} m_v{(l)} \cdots$,
'$m_v``$' has no nonlinear function whereas MemGAT has a nonlinear function for WRITE, .e.g,  $\alpha_{vv}^{(l)} \sigma^{(l)} (m_v^{(l)})$. Is this a typo? Otherwise, provide intuition why MemGCN should not use nonlinear activation functions for messages from memory nodes.
	2. Unlike memGCN, memGAT adjusts the effect of messages from memory nodes within the attention mechanism. Is the main reason why the MemGAT significantly underperforms MemGCN in table 2 even though the vanilla GAT consistently outperforms the vanilla GCN?

[1] Zhang, Songyang, Xuming He, and Shipeng Yan. "Latentgnn: Learning efficient non-local relations for visual recognition." International Conference on Machine Learning. 2019.

--- Post Rebuttal ---
I read the author feedback.  The typo in Question 1 is fixed and the issue with the edge weights is addressed. However, the proposed method requires model-specific modifications and cannot be applicable to other tasks on graphs, e.g., link prediction. Due to the limitations, I will keep the original rating.

---

> ### Author Response · Authors · 2020-11-22
> **Response**
>
> We thank the reviewer for providing comments. Below we address your specific points:
>
> (#1) On comparison with LatentGNN: LatentGNN is a technique to reduce the heavy computations (pairwise between all node features of the input graph) using latent nodes that are conceptually related to the memory nodes in our paper. However, we note that LatentGNN utilizes a different message passing protocol with that of message passing graph neural networks, in that message updates are indirectly transferred through latent nodes. In memGNNs, node messages are decomposed into messages from directly connected nodes and messages from the memory (latent) nodes. LatentGNN is suitable for situations where connectivity information is not available, i.e, computer vision scenarios. From this point of view, memGNN could also be viewed as a combination of locally unordered message passing with non-local message passing. We also include LatentGNN in the related work section, thanks for pointing out this line of work.
>
> (#2) On the need to modify backbone architectures: The effect of memory augmentation depends on the underlying tasks and dataset, for certain tasks memory-only enhancements may result in marginal improvements. However as we added the experiments with the PPI dataset in which we only add memory component with the rest setup remains almost the same with their backbone implementation, we are able to get significant improvements.
>
> (#3)On decoupling model performance with edge weights: We provide additional experiments that do not involve edge weights in the revised version of the paper, the results suggest that on actor and web datasets, the incorporation of memory component provides significant performance gain. For the citation dataset, we give an ablation study in the paper showing the gain of the memory component.
>
> On question 1: this is a typo and we fixed it in the revised version. Thanks for pointing out.

---

### Official Review · AnonReviewer3 · 2020-10-30
**Official Blind Review #3**

**Rating:** 3
**Confidence:** 4

**Review:**

This paper proposes to augment graph neural networks with awareness of global graph information based on memory neural networks.

While this work attempts to leverage the memory networks for latent representation transformation, so as to preserve global graph structural information, there exist many recent developed graph neural models which aims to inject global-level graph structure into the embedding generation. To name a few for reference:
Deep Graph Infomax, ICLR.
Position-aware Graph Neural Network, ICML.
However, the proposed models (MemGAT and MemGCN) are mainly compared to some representative graph neural network architectures, like GCN and GAT, which is insufficient to demonstrate the effectiveness of the new graph neural method.

As a general graph neural network model, the evaluation experiments on only the node classification task can hardly comprehensively justify the rationality of the proposed approach. Other benchmark tasks, such as link prediction or node clustering, could be considered in the evaluation section. Additionally, it would be interesting to discuss the latent influence between memory dimensions and graph network depth.

Another important dimension of evaluation lies in the model efficiency study. How is the computational cost of the proposed MemGNN (MemGCN and MemGAT) framework as compared to other alternatives, such as GCN and GAT, can be investigated. Due the pairwise relation learning with attention mechanism, the GAT-based neural architecture could be more time-consuming. What is the additional cost brought by the designed memory neural network with multi-dimensional latent representation projection.

From the evaluation results, we can observe that GAT- and GCN-based graph neural networks provide different performance with respect to split methods (e.g., random split and standard split). It would be interesting to provide some insights and clarifications to discuss this point, in order to understand the memory graph neural network better.

---

> ### Author Response · Authors · 2020-11-22
> **Response**
>
> We thank the reviewer for providing comments
>
> Additional tasks: Link prediction is indeed another interesting application that under suitable model definitions fits the memory augmentation framework. Due to time limits, we didn't include the study in the current revision.
>
> The additional complexity brought by the memory component: in terms of computation, the incorporation of memory component results in extra O(NM) message passings per layer. This extra cost could be considered large when the original graph is sparse. However since the goal of memory augmentation is to encode global information of the original graph, it seems to aggregate global information an additional O(N) computation is unavoidable.

---

### Author Response · Authors · 2020-11-22
**Revision**

We'd like to thank all the reviewers for providing helpful comments. We've made the following updates to our paper based on your feedback:

- To address concerns about the real-world existence of locally indistinguishable graphs, we add remark 1 to the paper that provides a discussion. In particular, strictly inditinguishable graphs are rarely seen in practice mainly due to the restriction that all nodes in the two graphs are mapped through a local isomorphism. If we relax the surjectivity constraints to a subset of nodes having their induced subtrees isomorphic, we get indistinguishable instances for finite layer LUMP GNNs. And such cases exist in bioinformatics as stated in the revised version of the paper

- On experimental setups excluding multi-graphs: we complement our experiments with an additional evaluation of MemGAT and MemGCN on the Protein-Protein Interaction (PPI) dataset, which contains 24 graphs. We obtained good results for both MemGAT and MemGCN which significantly improves the performance of their backbone architectures.

- On the effect of edge weights: to assess the effect of memory components in a more systematic way, we provide results without using edge weights on the actor and the web datasets. The results suggest that without edge weights, the incorporation of the memory component offers significant performance improvement.

Further, we respond to individual comments below.

---

### Decision · Program_Chairs · 2021-01-07
**Final Decision**

**Decision:**

Reject

**Comment:**

In this paper, the authors proposed a method to handle the problem of LUMP GNN architecture. This problem is indeed important and the proposed method has some merits. However, the proposed approach is only applicable to node classification.  Moreover, the proposed approach shows the similar theoretical results of Sato et al 2020. In the paper, it can be applicable to any GNN tasks because it only adds random features to each node. Therefore, the novelty of the proposed method is limited.  I encourage authors to revise the paper based on the reviewer's comments and resubmit it to a future venue.